# Saccharin and Sucralose Protect the Glomerular Microvasculature In Vitro against VEGF-Induced Permeability

**DOI:** 10.3390/nu13082746

**Published:** 2021-08-10

**Authors:** Emmanuella Enuwosa, Lata Gautam, Linda King, Havovi Chichger

**Affiliations:** 1Biomedical Research Group, Faculty of Science & Engineering, School of Life Sciences, Anglia Ruskin University, Cambridge CB1 1PT, UK; emmanuella_obi@yahoo.com (E.E.); Linda.King@aru.ac.uk (L.K.); 2Forensic Science Research Group, Faculty of Science & Engineering, School of Life Sciences, Anglia Ruskin University, Cambridge CB1 1PT, UK; lata.gautam@anglia.ac.uk

**Keywords:** artificial sweeteners, diabetic kidney disease, sweet taste receptor, glomerular, endothelium, vascular endothelial growth factor (VEGF), gas chromatograph-mass spectrometry (GC-MS), vascular permeability

## Abstract

Diabetic kidney disease (DKD) has become a global health concern, with about 40% of people living with type 1 and type 2 diabetes mellitus developing DKD. Upregulation of vascular endothelial growth factor (VEGF) in the kidney is a significant pathology of DKD associated with increased glomerular vascular permeability. To date, however, current anti-VEGF therapies have demonstrated limited success in treating DKD. Recent studies have shown that artificial sweeteners exhibit anti-VEGF potential. The aim of this study was therefore to assess the effects of aspartame, saccharin, and sucralose on VEGF-induced leak using an in vitro model of the glomerular endothelium. Saccharin and sucralose but not aspartame protected against VEGF-induced permeability. Whilst the sweeteners had no effect on traditional VEGF signalling, GC-MS analysis demonstrated that the sweetener sucralose was not able to enter the glomerular endothelial cell to exert the protective effect. Chemical and molecular inhibition studies demonstrated that sweetener-mediated protection of the glomerular endothelium against VEGF is dependent on the sweet taste receptor, T1R3. These studies demonstrate the potential for sweeteners to exert a protective effect against VEGF-induced increased permeability to maintain a healthy endothelium and protect against vascular leak in the glomerulus in settings of DKD.

## 1. Introduction

The increasing incidence of diseases linked to higher consumption of sugar-sweetened drinks has led to a shift towards a supposedly healthier option, such as low-calorie and non-nutritive sweeteners. Amongst these sweeteners are both synthetic (artificial) and naturally derived sweet-taste molecules, of which artificial sweeteners are more commonly found in both low-calorie and sugar-free foods and drinks [1]. The most commonly used, commercially available artificial sweeteners are acesulfame–K (Sunnett^®^, New York, NY, USA), aspartame (NutraSweet^®^, Augusta, GA, USA), saccharin (Sweet’N Low^®^, New York, NY, USA), and sucralose (Splenda^®^, Carmel, CA, USA) [2]. These sweeteners vary in their structure, intensity of sweetness, after-taste, and pharmacokinetics [1]. Artificial sweeteners elicit a distinctive perceived quality—termed “sweetness”—in humans through binding to the sweet taste receptor [3]. In humans, the sweet taste receptor belongs to the T1R class C of G protein-coupled receptors (GPCRs). The T1R has two subtypes: the T1R member 2 and 3 (T1R2 and T1R3), which form a heterodimer to act as a sweet taste receptor [4]. Whilst the T1R2/T1R3 is a heterodimer, the T1R3 subunit can also act as a homodimer to form a sweet taste sensor [3,5]. Indeed, the T1R3 subunit alone has been shown to play a vital role in sweet taste sensing in the vasculature [6,7,8]. The sweet taste receptor acts through the traditional GPCR-signalling pathway to activate the βγ G-protein subunits and phospholipase C-β (PLC-β) to stimulate cAMP production, triggering intracellular calcium release and resulting in sweet taste perception in the sweet taste buds in the mouth [9,10,11]. Of significance, the sweet taste receptor has been identified in a range of extraoral locations including the small intestine, heart, bladder, and bone [6,12,13,14]. Recently, the sweet taste receptor has also been identified in the vasculature of the lungs and retina, where it plays a key role in regulating endothelial barrier function and attenuating aberrant angiogenic processes [7,8]. These findings raise the question of whether the expression of sweet taste receptors occurs in other vascular beds and their potential impact on mechanisms related to vascular disruption. 

Diabetic kidney disease (DKD) is a form of microvascular complication and the major cause of end-stage renal disease in patients with diabetes [15,16,17]. The growing number of people with diabetes has a direct impact on DKD incidence, with approximately 30% and 40% of individuals with type 1 (T1DM) and type 2 (T2DM) diabetes mellitus respectively developing DKD over a median of 15 years from the time of disease diagnosis [18,19,20,21]. DKD is a progressive condition that begins with the thickening of the glomerular basement membrane, associated with loss of endothelial fenestrations and podocytes, followed by development of microaneurysms, leading to exudative lesions from subendothelial deposits of plasma proteins [22,23,24,25,26]. In later stages of DKD, interstitial changes and glomerulopathy can result in segmental and global sclerosis, associated with albuminuria [17]. Studies have shown that in DKD patients and rodent models of the disease, an increase in vascular endothelial growth factor (VEGF) has been associated with increased vascular permeability linked to hyperfiltration and albuminuria [27,28,29,30]. Given the pivotal role of glomerular vascular permeability in regulating DKD pathology, therapies for the disease have been focused on anti-VEGF approaches to attenuate this leak. Whilst effective in rodent models, a subset of diabetic patients treated with anti-VEGFs, such as Bevacizumab, display worsening of renal injury and excessive albuminuria [31,32]. There is, therefore, a need to develop vascular-specific therapeutics to reduce vascular permeability and therefore albuminuria in patients living with DKD without further exacerbating the disease. 

In the present study, our central hypothesis was that artificial sweeteners can protect the glomerular endothelium against VEGF-induced permeability. We proposed that this protective effect could be mediated by the sweet taste receptor, T1R3. We further sought to establish the molecular mechanism through which artificial sweeteners exert this protective effect, with a focus on cAMP-dependent signalling and oxidative stress in the endothelium. This study aimed to expand our understanding of the effect of sweeteners on the vasculature and potentially identify a new therapeutic option in the treatment of vascular disruption in DKD.

## 2. Materials and Methods

### 2.1. Cell Line and Reagents

Primary glomerular microvascular endothelial cells (GMVEC), purchased from Cell Systems (Kirkland, WA, USA), were cultured in complete classic medium (#4Z0-500) supplemented with culture boost. The passage number was used between 3 to 7. This primary cell line was utilised as any other relevant cell lines are immortalised rather than human primary cells and were therefore not appropriate. Endothelial growth factor (VEGF-A_165_) was purchased from Thermo-Fisher (Paisley, UK). Pure and analytical grade artificial sweeteners, aspartame, saccharin, and sucralose, the CCK-8 cell viability kit, FITC-dextran, *N*-Methyl-*N*-(trimethylsilyl) trifluoroacetamide (MSTFA), methanol, and ethyl acetate were purchased from Sigma-Aldrich (Dorset, UK). Lactisole, a sweet taste inhibitor, was purchased from Cayman Chemical (Ann Arbor, MI, USA). The cAMP-Screen Direct System kit and GSH Bioxytech activity kit were purchased from Applied Biosystems and Merck Millipore respectively. DharmaFECT™ reagent and siRNA (T1R3 and non-specific, scrambled) were purchased from Dharmacon (Cambridge, UK). Anti VE-cadherin and fluorescent secondary antibodies, deuterated sucralose, and sucralose-D6 (SC-220145) were purchased from Santa Cruz Biotechnology (Santa Cruz, CA, USA). 2,7-dichlorodihydrofluorescein diacetate (DCFDA) was purchased from Abcam (Cambridge, UK). 

### 2.2. Cell Viability Assay 

The effect of the artificial sweeteners on GMVEC viability was determined using the cell-counting kit-8 (CCK-8), following manufacturer’s guidelines. GMVEC were exposed to increasing concentrations (0.1–100 µM) of aspartame, saccharin, and sucralose separately and incubated for 24 h. A vehicle control of H_2_O was used for 0 µM. Absorbance was measured at 450 nm using a multi-mode microplate reader (Tecan Sunrise), and viability was calculated as % normalised to vehicle.

### 2.3. Endothelial Monolayer Permeability

Changes in endothelial monolayer permeability were assessed using FITC conjugated to 20 kDa dextran (FITC-D20). GMVEC were plated in a 24-well transwell plate (20,000 cells per well) and incubated for 24 h, followed by treatment with the named artificial sweeteners for 20 h at concentrations ranging from 0.1–100 µM or vehicle H_2_O control (0 µM). Cells were then exposed to VEGF (50 ng/mL) for a further 4 h. FITC-D20 was added to media in the upper chamber of the transwell filter at a concentration of 5 mg/mL and incubated for 180 s at 37 °C. Media from the lower and upper chamber (100 μL each) were then collected and analysed at excitation and emission wavelengths of 485 nm to 535 nm at 1 s exposure time using a fluorescent plate reader (Victor, Perkin Elmer). Permeability was calculated by fluorescence accumulated in the lower chamber divided by fluorescence remaining in the upper chamber. To validate the effect of VEGF on endothelial monolayer integrity, an EVOM^2^ meter (World Precision Instruments) was used to assess the trans-endothelial electric resistance (TEER) of the GMVEC monolayer in the presence and absence of VEGF (50 ng/mL) [8]. Experiments were repeated following siRNA knockdown of T1R3 using SMARTpool siGENOME siRNA duplexes (300 nM) or scrambled control duplexes with Dharmafect^TM^ reagent 4, as previously described [7,8]. 

### 2.4. ELISA Studies for VE-Cadherin and cAMP

For the investigation of VE-cadherin expression, following culture of GMVEC, the cells were exposed to aspartame, saccharin, or sucralose for 20 h at concentrations of 100 µM or vehicle H_2_O control (0 µM). Cells were then exposed to VEGF for a further 4 h. Following treatment, GMVEC were rinsed once with Dulbecco’s phosphate-buffered saline (DPBS) and fixed using 4% paraformaldehyde at room temperature for 10 min. An indirect whole cell ELISA was performed as previously described [7,8], using primary antibodies specific to the extracellular domain of VE-cadherin and fluorescent-conjugated secondary antibodies. Measurements at excitation and emission wavelengths of 495 nm to 520 nm were taken at 1 s exposure time using a florescent plate reader. Experiments were repeated following siRNA knockdown of T1R3 using SMARTpool siGENOME siRNA duplexes (300 nM) or scrambled control duplexes with Dharmafect^TM^ reagent 4, as previously described [7,8]. 

To assess the impact of sweeteners on the level of intracellular cyclic adenosine monophosphate (cAMP) in GMVEC, studies were performed following manufacturer’s guidelines for the cAMP-Screen Direct System. Cells were exposed to VEGF (50 ng/mL) or Forskolin (FSK), a broad activator of cAMP (10 µM), for 5, 15, and 30 min. Alternatively, GMVEC were exposed to aspartame (10 µM), sucralose (0.1 µM), and saccharin (0.1 µM) for 20 h followed by VEGF (50 ng/mL) exposure for a further 4 h followed by cAMP determination. Measurements were taken at 1 s exposure time (535/590 nm) using a florescent plate reader. 

### 2.5. Oxidative Stress Studies

To determine the protective effect of artificial sweeteners on the oxidative state of GMVEC, a known fluorogenic probe for the measurement of intracellular reactive oxygen species (ROS), 2,7-dichlorodihydrofluorescein diacetate (DCFDA), was utilised. For cellular glutathione (GSH) measurements, the GSH Bioxytech activity kit was used. GMVEC were seeded on black-walled, clear-bottomed, 96-well plates that had been pre-rinsed with an attachment factor and then incubated for 24 h. Following incubation, GMVEC were exposed to VEGF (50 ng/mL) or H_2_O_2_ (10 µM) for 4 h in the presence or absence of N-acetyl-cysteine (NAC). Either VEGF and NAC or H_2_O_2_ and NAC were added to cells at the same time. Alternatively, cells were exposed to aspartame (10 µM), saccharin (0.1 µM), or sucralose (0.1 µM) for 20 h followed by further 4 h exposure to VEGF (50 ng/mL). Irrespective of treatment, for ROS studies, DCFDA (10 µM) was then added to wells, and the level of ROS was measured on a fluorescent plate reader for 1 s at an excitation/emission of 485/535 nm. For GSH studies, levels of reduced or oxidised glutathione were measured on a fluorescent plate reader for 1 s at an excitation/emission of 380/461 nm. GSH levels were expressed as normalised to vehicle (H_2_O) treatment. 

### 2.6. Gas Chromatography–Mass Spectrometry

Gas Chromatography–Mass Spectrometry (GC–MS) analysis was adapted from Qiu et al. [33], using a Perkin Elmer Clarus 500 GC-MS equipped with an electron-impact ion source with an electron energy of 70 eV. The GC capillary column was an Rtx-5ms with a 30-m length, 0.25-mm internal diameter, and 0.25-µm thickness. The carrier gas was helium (BOC, 99.95%) at a flow rate of 0.6 mL/min. The instrumental parameters and method set-up were as follows: injection port temperature (270 °C); injection volume (1 µL); oven temperature parameters (180 °C for 2 min, 6 °C/min until 250 °C, hold for 20 min); transfer line temperature (280 °C); solvent delay time (3.5 min); injection mode (Split 9:1); and Scan 40–600 *m*/*z*. To optimise and validate the GC–MS method, selection of internal standard (sucralose-d6), linearity, limit of detection and quantification (LOD and LOQ), and precision were assessed following ICH guidelines [34]. This method was then applied to the cell studies. 

Sucralose (5 mg) powder was dissolved in 10 mL of methanol to make a 0.5-mg/mL (500 µg/mL) stock solution. The stock solution of sucralose was then diluted down with methanol to get the desired concentrations. To optimise the derivatising reaction of sucralose with MFSTA, preliminary studies included different reaction times (30, 45, 60 min) and temperature (room temperature and 70 °C). Based on the findings, the reaction temperature for 70 °C for 30 min was selected. Equal volumes (100 µL) of sucralose (100 µg/mL) and sucralose d6 (50 µg/mL) were aliquoted into a glass vial, and the solvent was evaporated using a miVac sample concentrator. Once the solvent was evaporated, 100 µL of MSTFA was added, and the mixture was heated at 70 °C for 30 min to derivatise. The derivatisation of sucralose using MSTFA led to the generation of sucralose-TMS, which was then injected into the GC–MS for analysis. To determine the suitability and stability of the internal standard, 50 µg/mL of sucralose-d6 was derivatised and analysed for 72 h. The analyte of interest, sucralose, in the presence of the internal standard, was analysed over 24 h for intraday precision and over 3 days to assess inter-day precision. Finally, calibration standards between 4–400 μg/mL were analysed and their LOD and LOQ calculated.

### 2.7. Preparation of GMVEC for GC–MS Analysis 

To determine the presence of sucralose in GMVEC, cultured cells were exposed to 4, 40, 100, 200 and 400 µg/mL sucralose concentrations for 24 h. GMVEC extraction was adapted from Gunduz et al. [35], excluding the derivatisation reaction. Following treatment with sucralose, used media was aspirated off, and cells were washed 6 times with phosphate-buffered saline (PBS). The GMVEC were then soaked in methanol and scraped. The cell suspension was transferred into tubes, sonicated, vortexed, and centrifuged at 1000 rpm for 10 min. The supernatant of the cell lysate was transferred into glass vials, mixed with 100 µL of internal standard, and dried. Derivatisation of the cell sample mix was done using 100 µL of MSTFA at 70 °C for 30 min and analysed utilising the GC–MS protocol for sucralose as outlined above. To determine whether the sucralose remained in the cell media after 24 h of post-sucralose treatment, the media from each well was removed and pipetted into glass vials. A total of 100 µL of the used media was mixed with 100 µL (50 µg/mL) of internal standard and dried and derivatised with 100 µL of MSTFA at 70 °C for 30 min, followed by GC–MS analysis. 

### 2.8. Statistical Analysis

All data sets were statistically analysed using GraphPad Prism version 8.2.0 for Windows (San Diego, CA, USA). Statistical analysis was performed using a one-way or two-way ANOVA with Tukey multiple comparisons post-hoc test where relevant. The experimental number is stated in the figure legend for each graph in the results section. The data is presented as mean ± standard error mean (S.E.M.) except where stated otherwise, with significance reached when *p* < 0.05.

## 3. Results

### 3.1. Artificial Sweeteners Saccharin and Sucralose Attenuate VEGF-Induced Permeability in Glomerular Microvascular Endothelium

The concentration of artificial sweeteners found in various foods and drinks differs according to each product consumed. However, there are specified acceptable daily intakes of these sweeteners: aspartame (40 mg/kg); saccharin (5 mg/kg); and sucralose (15 mg/kg) [2,36]. Bioavailability studies indicate concentration ranges of between 2–15% of the ingested dose in circulation, as the artificial sweeteners with the exception of aspartame are poorly absorbed [37,38,39]. Based on this evidence, a range of physiologically relevant concentrations of 0.1–100 µM were first assessed for any cytotoxic effect on GMVEC in vitro. The results showed that the artificial sweeteners, aspartame, saccharin, and sucralose, had no impact on GMVEC viability at concentrations of up to 100 µM (Figure 1a). 

The protective effect of aspartame, saccharin, and sucralose in maintaining the glomerular endothelial barrier was then investigated. VEGF was used to induce a significant increase in GMVEC monolayer permeability, as measured by FITC-dextran assay (Figure 1(bi)) and trans-endothelial electrical resistance (TEER-Figure 1(bii)). To assess the protective effect of sweeteners in settings of VEGF-induced leak, GMVEC were treated with sweeteners at varying doses, from 0.1–100 µM, in the presence and absence of VEGF. In the absence of VEGF, aspartame, saccharin, and sucralose exerted no effect on glomerular microvascular monolayer permeability (Figure 1c–e). Aspartame demonstrated a dose-selective effect, with protection against VEGF-induced permeability only observed at higher concentrations of 10 and 100 µM (Figure 1c). In contrast, saccharin and sucralose abolished VEGF-mediated monolayer leak at all the studied concentrations (Figure 1d,e). 

To further establish this protective effect of sweeteners on the glomerular endothelial barrier, the surface-level expression of the adherens junction protein, VE-cadherin, was investigated using the lowest barrier-protective concentration of each sweetener (aspartame—10 µM; saccharin and sucralose—0.1 µM). In the absence of VEGF, aspartame, saccharin, and sucralose maintained consistent surface-level expression of VE-cadherin in GMVEC (Figure 1f). Following exposure to VEGF, the sweeteners maintained high surface expression level of VE-cadherin at the cell surface, indicating a protected monolayer (Figure 1f). These findings demonstrate that the artificial sweeteners, aspartame, saccharin, and sucralose, are non-toxic to the endothelium and also attenuate VEGF-induced leak across the in vitro model of the glomerular microvasculature.

### 3.2. Aspartame, Saccharin, and Sucralose Attenuate VEGF-Induced Permeability via the T1R3 Sweet Taste Receptor

In humans, the sweet taste receptor belongs to the T1R class C of GPCRs, which is comprised of the T1R2 and T1R3 heterodimer [4]. Therefore, we next sought to assess whether the barrier-protective effect of the artificial sweeteners, aspartame, saccharin, and sucralose, was through T1R2/T1R3 activation. To address this, the chemical inhibitor of the sweet taste receptor, lactisole, was used [5,40]. Lactisole blocked the protective effect of all three sweeteners, aspartame (Figure 2a), saccharin (Figure 2b), and sucralose (Figure 2c), resulting in VEGF-induced permeability across the glomerular endothelium. Furthermore, glomerular endothelial cell-surface VE-cadherin expression demonstrated a similar pattern. Sweetener-induced protection against the loss of VE-cadherin at the GMVEC surface due to VEGF was blocked by lactisole (Figure 2d–f). 

We further studied the role of T1R3 in regulating sweetener-induced protection of the glomerular endothelium using T1R3-specific siRNA. Molecular inhibition of T1R3 had no impact on monolayer permeability (Table 1a) or VE-cadherin surface expression (Table 1b) of GMVEC in the absence of VEGF. However, in the presence of VEGF, T1R3 blocked the barrier-protective effect of aspartame, sucralose, and saccharin (Table 1a). This was mirrored by findings of VE-cadherin internalization, where T1R3 knockdown blocked the sweetener-induced maintenance of adherens junction protein noted in the presence of VEGF (Table 1b). 

Taken together, these results demonstrated that artificial sweeteners, aspartame, saccharin, and sucralose, protected the glomerular microvasculature against VEGF-induced barrier disruption in a sweet taste receptor-dependent manner.

### 3.3. Artificial Sweeteners Do Not Impact cAMP or Oxidative Stress Pathways in the Glomerular Microvasculature 

Our next experiments sought to determine the potential mechanism through which aspartame, saccharin, and sucralose protect the glomerular microvasculature against VEGF-induced leak. The most well-studied downstream signal of T1R2/T1R3 activation is the release of intracellular cyclic adenosine monophosphate (cAMP), a modulator of endothelial barrier function through reorganisation of the actin molecules and adherens junctional proteins associated with cell–cell adhesion [35,41]. Therefore, we next evaluated the effect of each artificial sweetener on intracellular cAMP levels in the glomerular microvasculature. We first established the cAMP assay using Forskolin, which activates adenyl cyclase (Figure 3a), and VEGF as our model of vascular injury (Figure 3b). Both increased intracellular cAMP levels significantly from 5–15 min in a time-dependent manner. Interestingly, in the presence and absence of VEGF, the artificial sweeteners, aspartame, sucralose, and saccharin, had no effect on intracellular cAMP levels in GMVEC (Figure 3c). 

We next studied the effect of the artificial sweeteners, aspartame, saccharin, and sucralose, on the endothelial stress signal, reactive oxygen species (ROS). Intracellular accumulation of ROS is linked to tyrosine phosphorylation of VE-cadherin and the subsequent loss of cell–cell adhesion, resulting in vascular leak [42,43]. We utilised an established probe for the measurement of intracellular ROS, 2,7-dichlorodihydrofluorescein diacetate (DCFDA), and demonstrated the probe sensitivity using hydrogen peroxide (H_2_O_2_) to induce ROS and the antioxidant N-acetyl cysteine (NAC) to decrease ROS (Figure 3d). In the absence of VEGF, aspartame significantly increased ROS accumulation in GMVEC, whilst saccharin and sucralose had no effect (Figure 3e). In the presence of VEGF, there was a significant increase in ROS accumulation, which was unaffected by exposure to saccharin, sucralose, and aspartame (Figure 3e). This finding was mirrored in studies assessing cellular glutathione (GSH) levels in GMVEC, with VEGF significantly lowering GSH expression and sweeteners having no impact on the oxidative process (Figure 3f). 

Taken together, these findings demonstrate that artificial sweeteners, aspartame, saccharin, and sucralose, act through a non-traditional signalling pathway in the glomerular microvasculature. 

### 3.4. Analysis and Detection of Sucralose by GC–MS

To further understand how sweeteners impact the endothelium and given that T1R2/T1R3 endocytosis has been previously demonstrated [44], we next evaluated the trafficking of artificial sweeteners into the endothelial cell. To address this, we utilised an analytical technique, GC–MS, and focused on the sweetener that has been studied using this technique, sucralose [45]. Polar compounds, such as sucralose, require chemical modification, known as derivatisation, prior to analysis by GC-MS [45]. Following derivatisation of sucralose with MSTFA, the hydroxyl functional groups of sucralose were replaced with a trimethyl silyl group. Derivatised sucralose (sucralose-TMS) was detected with a peak at 22.29 min retention time and the internal standard (sucralose-d6) at 22.14 min (Figure 4). The identification of the peak as sucralose-TMS was further confirmed through the examination of its mass spectra. The main ions of interest of sucralose were at mass/charge (*m*/*z*) 207, 308, and 343 (Figure 5b) [33], differentiating it from those of the internal standard (Figure 5a), whose main ions were at *m*/*z* 211, 312, and 347. 

Prior to its application to cell studies, the GC–MS method was optimised and validated. The suitability of internal standards, precision, linearity, limit of detection (LOD), and limit of quantification (LOQ) were assessed [46]. The stability of the sucralose-d6 over 72 h (Figure 6a) demonstrated its suitability as an internal standard (RSD < 5%). Due to co-elution of the internal standard, an extracted ion chromatogram (sucralose: *m*/*z* 308; sucralose-d6: *m*/*z* 312) was used for data analysis. Our results from inter- and intra-day precision studies further show (Figure 6b,c) an RSD below 5%, meeting the Scientific Working Group for Forensic Toxicology (SWGTOX) and other international guidelines [46,47]. The regression plot of sucralose was linear from 4–400 µg/mL (r^2^ = 1.000), and detection and quantification limits were calculated as 0.2-ng and 0.6-ng mass on column, respectively. 

### 3.5. The Artificial Sweetener Sucralose Does Not Enter the Glomerular Endothelium

Finally, we sought to determine whether sucralose is transported into and across the glomerular microvascular endothelium as a potential mechanism of attenuating VEGF-induced permeability. Following the optimisation and validation of the developed method for GC–MS detection of sucralose (Figure 4 and Figure 5), the same technique was applied to cell lysates. Our results indicated that no sucralose was detected in the cytosol of glomerular microvascular endothelial cells but rather remained in the extracellular space (cell media) (Figure 6d,e). These results, therefore, suggests that the artificial sweetener, sucralose, is not endocytosed nor diffuses into the glomerular microvascular endothelial cell and therefore is likely to protect against VEGF-induced permeability through an alternative mechanism. 

## 4. Discussion

The aim of the present study was to determine the protective effect of artificial sweeteners, aspartame, saccharin, and sucralose, on VEGF-induced permeability in a cell model of the glomerular microvascular endothelium. We demonstrated that the artificial sweeteners displayed differential effects on GMVEC processes. Whilst saccharin and sucralose protected the in-vitro glomerular microvasculature model against VEGF-induced leak at all studied concentrations, aspartame was only protective at higher concentrations. Chemical inhibition of T1R2/T1R3 and molecular inhibition of T1R3 blunted this protective effect; however, we demonstrated that the sweeteners did not impact cAMP levels, a traditional downstream signalling second messenger of T1R2/T1R3, or ROS accumulation, a typical regulator of endothelial cell health. Lastly, we developed a novel analytical tool to study sucralose levels and used this to demonstrate that the sweetener cannot enter the glomerular endothelial cell via T1R2/T1R3 endocytosis to exert this protective effect. Overall, these studies indicate the potential role of sweeteners as a novel therapeutic option in ameliorating glomerular vascular leak observed in individuals living with DKD. 

In the glomerulus, VEGF released from podocytes plays a key role in promoting endothelial fenestration integrity and regulating barrier function [48]. However, excessive VEGF release has been associated with diabetic kidney disease [27,49]. Furthermore, VEGF levels positively correlate with albuminuria in patients with diabetes, suggesting that the growth factor plays a key role in renal injury [50]. Overstimulation of the VEGF receptor 2 (VEGFR2) in a genetic mouse model results in mesangial matrix expansion, endothelial cell proliferation, and massive proteinuria [51]. There is also an increase in capillary number and area in the early stages of DKD, indicative of excessive angiogenesis, associated with vascular leak and regrowth [52]. As such, anti-VEGF therapies have been studied in animal models of both type 1 and type 2 diabetes with some success observed; for example, decreased albuminuria and reduced glomerular hypertrophy have been observed [30,53,54]. However, in clinical studies using the humanized monoclonal antibody, bevacizumab, proteinuria, and hypertension have been reported, linked to swelling of the glomerular endothelial cells and thrombotic microangiopathy [55,56,57]. There are, therefore, potential therapeutic applications for an anti-VEGF therapy for patients with DKD; however, there is a need to overcome the associated side effects and worsening of renal injury. 

In the present study, we investigated a novel regulator of glomerular endothelial function, namely artificial sweeteners. We demonstrated that aspartame, saccharin, and sucralose attenuated VEGF-induced leak; however, in the absence of VEGF, these sweeteners exerted no negative effect on barrier function or cytotoxicity. These studies thus indicate that sweeteners will not impact the healthy endothelium but will protect the leaky endothelium and therefore could offer a potential for use as a therapeutic option for patients with DKD. However, it is worth noting that aspartame increased oxidative stress in GMVEC in the absence of VEGF. These findings are consistent with our previous studies, which demonstrated a barrier-protective effect of the artificial sweetener, sucralose, in settings of VEGF-induced retinopathy and LPS-induced pulmonary edema [7,8]. Furthermore, Schiano et al. identified a range of genes in endothelial cells that are affected by sweeteners, including *CDH5*, *NOS3*, *HIF1A,* and *PTK2* [58]. The products of these genes are vital in reducing oxidative stress and improving barrier integrity in the microvasculature. However, in our present study, we showed that saccharin, sucralose and aspartame have no effect on oxidative stress in GMVEC, but they did tighten the microvascular barrier. It is therefore likely that sweeteners protect the endothelium through a ROS-independent mechanism. Given our findings showing that sweeteners stabilised VE-cadherin expression at the cell surface in settings of VEGF exposure, it is possible that they can directly impact adherens junction stability to protect against VEGF-induced barrier disruption. Interestingly, we also demonstrated that the barrier-protective effect of these sweeteners was blocked by exposure to the T1R2/T1R3 inhibitor, lactisole, and molecular inhibition of T1R3 had the same effect. These studies indicated that activation of T1R3 is essential in exerting the barrier-protective effect of sweeteners. Previous studies have shown the presence of T1R2/T1R3 in the kidney or indicate the importance of T1R3 in regulating other vascular beds [7,8,59]; however, the present work is the first to study the role of sweeteners and the sweet taste receptor in regulating vascular function in the glomerulus. 

Whilst all three sweeteners studied were protective against VEGF-induced leak across the glomerular endothelium, both saccharin and sucralose were protective at low concentrations (0.1 µM), whilst aspartame was only protective at higher concentrations (>10 µM). This may be due to the higher binding affinity of saccharin and sucralose sweetener to the T1R2/T1R3 complex as compared to aspartame [60] such that a higher concentration of aspartame is needed to stimulate the sweet taste receptor and promote barrier protection against VEGF injury. Despite this difference, it is worth noting that the concentration range of sweeteners that was studied here, 0.1 to 100 µM, is within the normal range that can be found in circulation [37,38,39]. In theory, it is therefore possible that consumption of artificial sweeteners in the diet could improve vascular function in patients with DKD. However, there are many studies that indicate the worsening of glucose intolerance, disruption of gut function, and vascular reactivity following consumption of sucralose, saccharin, and acesulfame K [14,61,62]; therefore, further study is needed to establish how to harness the barrier-protective role of artificial sweeteners without worsening the health of patients with diabetes. 

Following activation by a range of sweet taste molecules, including artificial sweeteners, the downstream signalling of T1R2/T1R3 typically involves cAMP-dependent activation of PLC [10,63]. In the present study, we observed no effect of sweeteners on intracellular cAMP levels in the presence or absence of VEGF. Whilst we demonstrated the efficacy of the assay to evaluate cAMP using both Forskolin and VEGF, it is possible that the sweeteners regulate local or compartmentalised cAMP levels, which cannot be evaluated with this ELISA. However, it is also possible that the sweeteners do not act through a traditional taste-signalling pathway in the glomerular endothelium. Previous studies have indicated that desensitisation of the T1R2/T1R3 occurs through endocytosis; however, it is not clear whether this internalisation process includes sweeteners bound to T1R2/T1R3 [44]. Therefore, to further understand the impact of sweeteners on the glomerular endothelium, we developed a sensitive analytical tool to detect sweeteners in the endothelial cell, with a focus on sucralose as the more well-studied sweetener in the microvasculature [7,8]. Our findings conclude that sucralose is not internalised into the glomerular endothelial cell cytosol, which indicates that further study is needed to understand the molecular mechanism through which sucralose and saccharin exert an antioxidant and barrier-protective role on the endothelium. 

## Figures and Tables

**Figure 1 nutrients-13-02746-f001:**
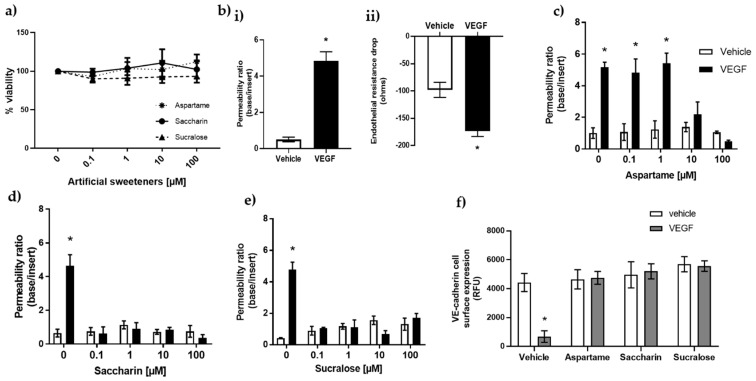
**Artificial sweeteners, saccharin and sucralose, attenuate VEGF-induced leak across the glomerular microvasculature.** Panel (**a**): viability of GMVEC, determined using a CCK-8 assay following exposure to artificial sweeteners, aspartame, saccharin, and sucralose, at physiologically relevant concentrations of 0.1–100 µM. Panel (**b**): effect of VEGF (50 ng/mL, 4 h) on permeability of GMVEC monolayers measured with FITC-dextran assay panel (**bi**) and validated by TEER panel (**bii**). Panel (**c**–**e**): protection of GMVEC monolayer following exposure to aspartame (**c**), saccharin (**d**), and sucralose (**e**) against VEGF-induced leak. Panel (**f**): effect of aspartame, saccharin, and sucralose on surface-level expression of VE-cadherin in the presence and absence of VEGF. Data are presented as mean with S.E.M and *n* = 6. * *p* < 0.05 vs. vehicle.

**Figure 2 nutrients-13-02746-f002:**
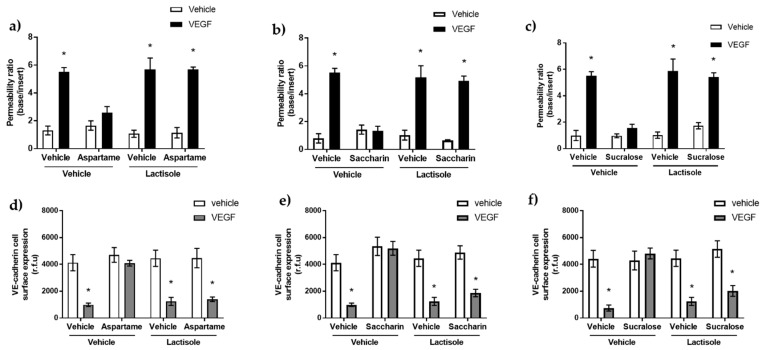
Inhibition of T1R3 using lactisole blocked sweetener-induced protection against VEGF-induced barrier leak across the glomerular microvascular endothelial cell. Panel (**a**–**c**): Inhibition of sweet taste receptors by lactisole blocks the protective effect of aspartame, saccharin, and sucralose. GMVEC were exposed to 10 µM aspartame (**a**), 0.1 µM saccharin (**b**), and 0.1 µM sucralose (**c**) in the presence and absence of VEGF (50 ng/mL) after pre-treatment with lactisole for 10 min, and permeability was assessed using FITC-D20. Panel (**d**–**f**): Whole cell indirect ELISA was used to determine GMVEC cell-surface expression of VE-cadherin following exposure to aspartame (**d**), saccharin (**e**), sucralose (panel f), and vehicle (media) in the presence and absence of VEGF and 3 µM of sweet taste inhibitor, lactisole. *n* = 6. Data are expressed as mean with S.E.M * *p* < 0.05 vs. vehicle.

**Figure 3 nutrients-13-02746-f003:**
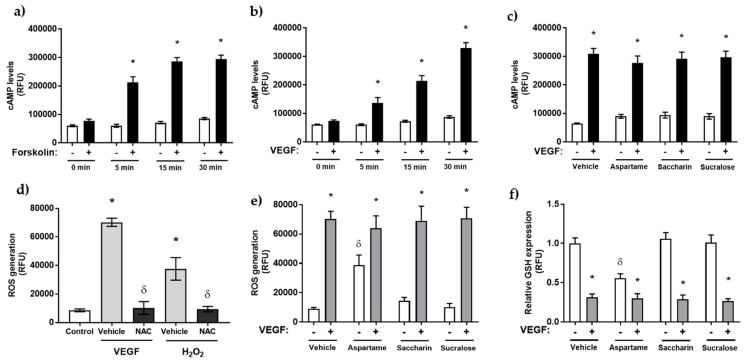
Artificial sweetener regulation of the glomerular endothelium is independent of cAMP and intracellular ROS generation. Panel (**a**–**c**): generation of intracellular cAMP upon exposure of GMVEC to aspartame (10 µM), saccharin (0.1 µM), and sucralose (0.1 µM) for 24 h in the presence and absence of VEGF (panel **b**) using the cAMP-Screen Direct chemiluminescent ELISA assay. Forskolin (FSK), 10 µM, was used to validate the assay (panel **a**). Panel (**d**–**f**): GMVEC were treated with VEGF or H_2_O_2_ (10 µM) or vehicle (H_2_O) in the presence or absence of NAC (10 µM), followed by incubation with DCFDA (10 µM) and fluorescence analysis (panel **d**). These experiments were repeated following pre-exposure to aspartame (10 µM), saccharin (0.1 µM), and sucralose (0.1 µM) for 2 h in the presence and absence of VEGF (panel **e**). ROS production was assessed by measuring the florescence level of DCF following staining with 10 µM DCFDA. For cellular glutathione (GSH) measurements (panel **f**), GMVEC were pre-exposed to aspartame (10 µM), saccharin (0.1 µM), and sucralose (0.1 µM) in the presence and absence of VEGF. Data are expressed as mean with S.E.M and *n* = 6. * *p* <0.05 vs. vehicle for sweetener; ^δ^
*p* < 0.05 vs. vehicle for VEGF or H_2_O_2_.

**Figure 4 nutrients-13-02746-f004:**
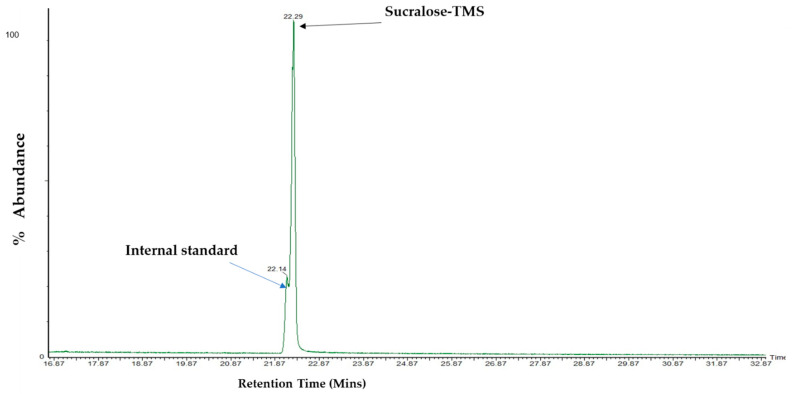
**Chromatogram of sucralose-TMS and its internal standard.** The chromatogram was recorded with a Perkin Elmer Clarus 500 GC–MS. GC–MS conditions: column Rtx ^®^-5 (30 m × 0.25 μm × 0.25 mm i.d.); 270 °C injection port temperature; column temperature: 180 °C for 2 min, 6 °C/minute until 250 °C, hold for 20 min with 3.5 min solvent delay. Arrow (black) shows peak detection of sucralose-TMS, which elutes at a retention time of 22.2 min, while the internal standard (blue arrow) was eluted at a retention time of 22.1 min.

**Figure 5 nutrients-13-02746-f005:**
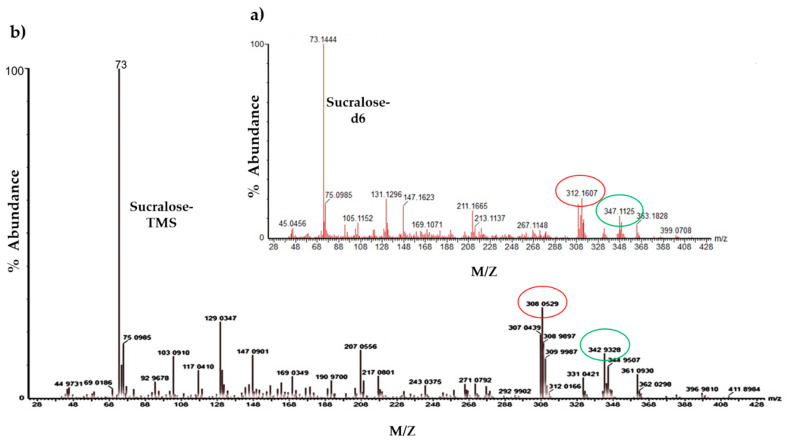
**Mass spectra for derivatised/silylated** (**a**) **sucralose-d6 and** (**b**) **sucralose.** The mass spectrum (mass/charge) for the internal standard, sucralose-d6 (panel **a**), and sucralose-TMS (panel **b**). The base ions of both mass spectra (circled red) are *m*/*z* 312 for internal standard and *m*/*z* 308 for sucralose-TMS. The qualifier ions (circled green) are *m*/*z* 347 for the internal standard and *m*/*z* 343 for sucralose-TMS following GC–MS analysis.

**Figure 6 nutrients-13-02746-f006:**
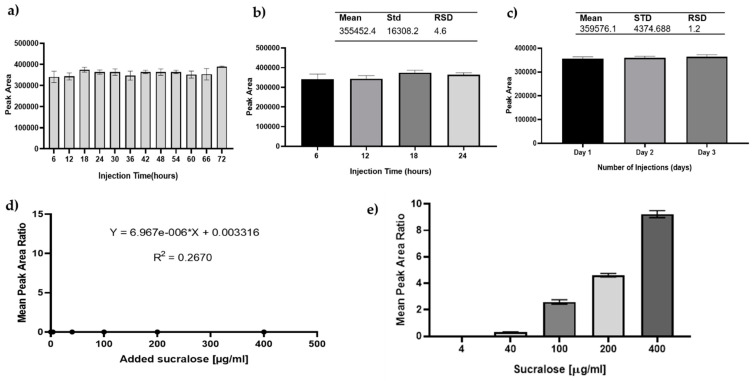
**Sucralose was not taken into glomerular endothelial cells.** Instrument auto-sampler stability was evaluated as part of the method validation prior cell analysis. Panel (**a**): bar graph of peak area of internal standard (50 µg/mL) plotted against various injection times to determine the autosampler stability of the instrument (GC–MS) and suitability of the internal standard. There was no statistically significant difference between the injections and control using two-way ANOVA. Panel (**b**,**c**): inter and intra-day precision of sucralose-d6. The intra-day (panel **b**) covers data generated over 24 h, while inter-day (panel **c**) covers an analysis time of 3 days. The results were analysed using two-way ANOVA and error bars presented as standard deviation with 36 technical repeats. The calculated RSD shown on the top of the bars shows RSD deviation between repeated runs. Results in panels (**d**) and (**e**) indicate that sucralose is not transported across the glomerular endothelial cell membrane. GMVEC were plated and incubated at 37 °C for 24 h, followed by exposure to 500 µL sterile-filtered sucralose concentrations ranging from 4 to 400 µg/mL and then incubated for an additional 24 h. The cell lysate (panel **d**) and used media (panel **e**) were aliquoted into glass vials, mixed with internal standard, evaporated to dryness, derivatised with MSTFA at 70 °C for 30 min, and analysed using GC–MS. The instrument (GC) conditions were as outlined in the methods section. The mean peak area ratio of the mixed analyte was plotted against sucralose concentrations. *n* = 3 of independent experimental repeats.

**Table 1 nutrients-13-02746-t001:** Molecular inhibition of T1R3 with siRNA has no impact on sweetener-induced protection against VEGF-induced barrier leak across the glomerular microvascular endothelial cell. GMVEC were transiently transfected with T1R3 siRNA or non-specific scrambled control siRNA (300 nM) for 24 h followed by exposure to artificial sweeteners aspartame (10 µM), saccharin (0.1 µM), and sucralose (0.1 µM) for 20 h. Cells were then exposed to VEGF (50 ng/mL) for a further 4 h. Permeability was assessed using FITC-dextran assay (panel a), and VE-cadherin cell-surface expression was measured using the whole-cell ELISA (panel b). *n* = 5–6. Data are expressed as mean with S.E.M * *p* < 0.05 vs. vehicle for VEGF.

(**a**)								
**siRNA**	**Permeability Ratio (Base/Insert)**							
	**Scrambled**				**T1R3**			
**Treatment:**	**Vehicle**	**Aspartame**	**Saccharin**	**Sucralose**	**Vehicle**	**Aspartame**	**Saccharin**	**Sucralose**
Vehicle	1.77 ± 0.25	1.99 ± 0.57	1.88 ± 0.73	1.71 ± 0.58	1.95 ± 0.38	2.06 ± 0.61	1.89 ± 0.67	1.59 ± 0.73
VEGF	5.91 ± 0.62 *	2.05 ± 0.41	2.01 ± 0.60	1.89 ± 0.36	5.79 ± 0.55 *	1.83 ± 0.71	2.13 ± 0.82	1.98 ± 0.40
(**b**)								
**siRNA**	**VE-Cadherin Cell-Surface Expression (r.f.u.)**							
	**Scrambled**				**T1R3**			
**Treatment:**	**Vehicle**	**Aspartame**	**Saccharin**	**Sucralose**	**Vehicle**	**Aspartame**	**Saccharin**	**Sucralose**
Vehicle	4038 ± 693	4126 ± 827	4150 ± 1017	4061 ± 904	4183 ± 1027	5199 ± 757	4289 ± 863	4890 ± 1007
VEGF	971 ± 183 *	4090 ± 938	4281 ± 1105	4135 ± 1036	891 ± 130 *	4937 ± 812	3980 ± 1170	4136 ± 1021

## Data Availability

Not applicable.

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
