# Peer review of "Saccharin and Sucralose Protect the Glomerular Microvasculature In Vitro against VEGF-Induced Permeability"

_nutrients, 2021, doi:10.3390/nu13082746_

Round 1
Reviewer 1 Report
Dear Authors
Title
The Title "Artificial sweeteners, saccharin and sucralose, protect the glomerular microvasculature against VEGF-induced permeability" should be correct since the current reader may have a different perception of the article impact through the linguistic option presented (and mainly because of the comma ",") by the authors which lead to interpretation of possible effect of all artificial sweeteners which is not the case. Of course the reader that reads all the paper understands that misinterpretation, but for unequivocal message I recommend the tithe should be changed for example as follows:
"Saccharin and sucralose artificial sweeteners protect the glomerular microvasculature against VEGF-induced permeability"
Also,
from an ethical point of view the correct message would be:
"Artificial sweeteners saccharin and sucralose protected in vitro the
glomerular microvasculature against VEGF-induced permeability using an in vitro model of
the glomerular endothelium"
Also, Subtitle "The artificial sweetener, sucralose, does not enter the glomerular endothelium" (line 387) should not have both the comma "," as this was even the criteria for other subtitles which are perfect from linguistic perspective
Gas Chromatography- Mass Spectrometry
The "derivatization" is not clear. You have used deuterated sucralose. Was it prepared in the lab? The you have written: The derivatising conditions for sucralose, reaction time and temperature were optimised. But this seems to be related with current MSTFA deivatization.... "Once the solvent was evaporated, 100 µl of 191
MSTFA was added and the mixture was heated at 70ºC for 30 minutes to derivatise, followed by injections into the GC-MS for analysis." ... that should lead to sucralose-TMS? What else have you done as "derivatization". Please clarify this paragraph.
Best regards,
Author Response
Reviewer 1
- The Title "Artificial sweeteners, saccharin and sucralose, protect the glomerular microvasculature against VEGF-induced permeability" should be correct since the current reader may have a different perception of the article impact through the linguistic option presented (and mainly because of the comma ",") by the authors which lead to interpretation of possible effect of all artificial sweeteners which is not the case. Of course, the reader that reads all the paper understands that misinterpretation, but for unequivocal message I recommend the tithe should be changed for example as follows: "Saccharin and sucralose artificial sweeteners protect the glomerular microvasculature against VEGF-induced permeability"
- Also, from an ethical point of view the correct message would be: "Artificial sweeteners saccharin and sucralose protected in vitrothe glomerular microvasculature against VEGF-induced permeability using an in vitro model of the glomerular endothelium"
We thank the reviewer for these comments and agree with the need to include the term ‘in vitro’ in the title – we have done so in the revised manuscript. We do, however, respectfully disagree with the reviewer regarding the phrasing of the title. We believe that it would be grammatically incorrect to state “Saccharin and sucralose artificial sweeteners protect….”. However, we appreciate the source of confusion and have adjusted to “The artificial sweeteners saccharin and sucralose protect….”.
- Also, subtitle "The artificial sweetener, sucralose, does not enter the glomerular endothelium" (line 387) should not have both the comma "," as this was even the criteria for other subtitles which are perfect from linguistic perspective.
We have adjusted this subtitle accordingly.
- Gas Chromatography- Mass Spectrometry. The "derivatization" is not clear. You have used deuterated sucralose. Was it prepared in the lab? The you have written: The derivatising conditions for sucralose, reaction time and temperature were optimised. But this seems to be related with current MSTFA derivatization.... "Once the solvent was evaporated, 100 µl of MSTFA was added and the mixture was heated at 70ºC for 30 minutes to derivatise, followed by injections into the GC-MS for analysis." ... that should lead to sucralose-TMS? What else have you done as "derivatization". Please clarify this paragraph.
We thank the reviewer for this comment. Commercially-available deuterated sucralose (sucralose-d6) was used in this research. We have included manufacturer’s details in the ‘Cell lines and reagents’ subsection of Methods (lines 89-90).
- The derivatising conditions for sucralose, reaction time and temperature were optimised. But this seems to be related with current MSTFA derivatization.... "Once the solvent was evaporated, 100 µl of MSTFA was added and the mixture was heated at 70ºC for 30 minutes to derivatise, followed by injections into the GC-MS for analysis." ... that should lead to sucralose-TMS? What else have you done as "derivatization". Please clarify this paragraph."
We thank the reviewer for this comment and agree with the need to clarify the paragraph. Preliminary studies were done on the derivatisation of sucralose with MSTFA at different times (30, 45 and 60 minutes) and temperature (room temperature and at 70 0C) to determine the optimal derivatisation parameters. Based on those preliminary assays, the reaction temperature of 70ºC for 30 minutes was selected. Data from preliminary findings was not included in the manuscript as this is not our focus of the paper. We have adjusted the Methods (lines 160-163) to reflect this.
Reviewer 2 Report
Material and Methods
Please, specify which kind of “classic medium” is used for the Primary glomerular microvascular endothelial cells (GMVEC) culture.
“cell counting kit-8 (CCK-8), following manufacturer’s guidelines” please add the manufacture
“GMVEC were plated in a 24-well transwell plate and incubated for 24 hours” please, the author should specify how many cells were are seeded. Moreover, the authors incubated the cells for 20 hours with H2O2, which is the concentration of H2O2 they used? It is well known that H2O2 is deleterious for cells, it is surprising that after 20 hours the cells are still alive.
The TEER experiment should be performed also before the treatments to be sure of the presence of the monolayer. Is this test performed?
Please add the manufacture of 2,7-dichlorodihydrofluorescein diacetate (DCFDA) and of GSH Bioxytech.
“Following incubation, GMVEC were exposed to VEGF (50 ng/ml) or H2O2 (10 μM) for 4 hours in the presence or absence of N-acetyl-cysteine (NAC).” It is not clear when NAC treatment was performed, before or after VEGF and H2O2 treatment?
CONCLUSION
This study could be accepted with major revision.
This study demonstrated the beneficial effects of artificial sweeteners on Primary glomerular microvascular endothelial cells (GMVEC) permeability. Indeed, it is well demonstrated that their treatments restored the permeability also in the presence of VEGF. Moreover, the treatment with the aspartame induced ROS production, however, in the presence of VEGF all the sweeteners treatments can’t inhibit ROS production: figure 3e, in the presence of VEGF, the increase of ROS is still significant vs CTR; thus, the sentence in the discussion: “Interestingly, saccharin and sucralose, but not aspartame, attenuated VEGF-induced ROS accumulation.” should be revised. Please, the authors has to clarify this point: how the sweeteners can restore the endothelial permeability, marker of endothelial healthy, even if they induced ROS production?.
Author Response
Reviewer 2
- Material and Methods: Please, specify which kind of “classic medium” is used for the Primary glomerular microvascular endothelial cells (GMVEC) culture.
This is a proprietary media so the specific content is not available however we have included the product code for this item in the text (line 80) to provide detail for the reader.
- “cell counting kit-8 (CCK-8), following manufacturer’s guidelines” please add the manufacturer.
We thank the reviewer for this comment however we have already included the manufacturer for the CCK-8 kit in the subsection titled ‘Cell lines and reagents’ lines 83-85.
- “GMVEC were plated in a 24-well transwell plate and incubated for 24 hours” - please, the author should specify how many cells were are seeded. Moreover, the authors incubated the cells for 20 hours with H2O2, which is the concentration of H2O2 they used? It is well known that H2O2 is deleterious for cells, it is surprising that after 20 hours the cells are still alive.
We thank the reviewer for this comment. We used 20,000 cells per well for transwell studies and we have now included this in the manuscript (line 102). To clarify, cells were incubated for 20 hours with H2O and not H2O2 for studies. H2O was the vehicle for the sweeteners studied and used as a negative control. We used a concentration of 10 µM of H2O2 (line 136) and exposed cells for 4 hours which is sufficient to cause oxidative stress without cell death as previously established (for example, Yamamura H et al, 2020, Biochem Biophyc Res Commun, 523(1); 153).
- The TEER experiment should be performed also before the treatments to be sure of the presence of the monolayer. Is this test performed?
Yes, we have performed these studies prior to treatments and typically see TEER values -118 ± 26 ohms. We did not include this in the manuscript as, following review of similar publications, this is not typically included.
- Please add the manufacturer of 2,7-dichlorodihydrofluorescein diacetate (DCFDA) and of GSH Bioxytech.
We thank the reviewer for this comment however we have already included the manufacturer details in the subsection titled ‘Cell lines and reagents’ – GSH Bioxytech kit lines 86-87 and DCFDA lines 90-91.
- “Following incubation, GMVEC were exposed to VEGF (50 ng/ml) or H2O2 (10 μM) for 4 hours in the presence or absence of N-acetyl-cysteine (NAC).” It is not clear when NAC treatment was performed, before or after VEGF and H2O2 treatment?
We added NAC with either VEGF or H2O2 and have now included a statement to explain this (lines 137-138).
- CONCLUSION: This study could be accepted with major revision. This study demonstrated the beneficial effects of artificial sweeteners on primary glomerular microvascular endothelial cells (GMVEC) permeability. Indeed, it is well demonstrated that their treatments restored the permeability also in the presence of VEGF. Moreover, the treatment with the aspartame induced ROS production, however, in the presence of VEGF all the sweeteners treatments can’t inhibit ROS production: figure 3e, in the presence of VEGF, the increase of ROS is still significant vsCTR; thus, the sentence in the discussion: “Interestingly, saccharin and sucralose, but not aspartame, attenuated VEGF-induced ROS accumulation.” should be revised.
Please, the authors have to clarify this point: how can the sweeteners restore the endothelial permeability, marker of endothelial healthy, even if they induced ROS production?
We appreciate the reviewer noting this inconsistency and have corrected the section highlighted in the results (lines 271-288) to correctly state that sweeteners have no effect on VEGF-induced ROS accumulation or VEGF-induced reduced GSH expression. That is, they do not increase or decrease oxidative stress in the endothelial cell and therefore are not a potential mechanism through which sweeteners could protect the endothelial barrier. We have addressed this in the discussion (lines 383-388) to expand on and clarify the point.
Round 2
Reviewer 2 Report
Thank to the authors for their comments.
I think that no more revisions are required.